# Mechanisms, Diagnosis and Treatment of Bone Metastases

**DOI:** 10.3390/cells10112944

**Published:** 2021-10-29

**Authors:** Jozef Ban, Valerie Fock, Dave N. T. Aryee, Heinrich Kovar

**Affiliations:** 1St. Anna Children’s Cancer Research Institute, 1090 Vienna, Austria; jozef.ban@ccri.at (J.B.); valerie.fock@ccri.at (V.F.); dave.aryee@ccri.at (D.N.T.A.); 2Department of Pediatrics, Medical University Vienna, 1090 Vienna, Austria

**Keywords:** bone metastasis, metastatic niche, tumor microenvironment interactions, bone colonization, EMT, metastatic dormancy, bone reconstruction, metastasis targeted therapy

## Abstract

Bone and bone marrow are among the most frequent metastatic sites of cancer. The occurrence of bone metastasis is frequently associated with a dismal disease outcome. The prevention and therapy of bone metastases is a priority in the treatment of cancer patients. However, current therapeutic options for patients with bone metastatic disease are limited in efficacy and associated with increased morbidity. Therefore, most current therapies are mainly palliative in nature. A better understanding of the underlying molecular pathways of the bone metastatic process is warranted to develop novel, well-tolerated and more successful treatments for a significant improvement of patients’ quality of life and disease outcome. In this review, we provide comparative mechanistic insights into the bone metastatic process of various solid tumors, including pediatric cancers. We also highlight current and innovative approaches to biologically targeted therapy and immunotherapy. In particular, we discuss the role of the bone marrow microenvironment in the attraction, homing, dormancy and outgrowth of metastatic tumor cells and the ensuing therapeutic implications. Multiple signaling pathways have been described to contribute to metastatic spread to the bone of specific cancer entities, with most knowledge derived from the study of breast and prostate cancer. However, it is likely that similar mechanisms are involved in different types of cancer, including multiple myeloma, primary bone sarcomas and neuroblastoma. The metastatic rate-limiting interaction of tumor cells with the various cellular and noncellular components of the bone-marrow niche provides attractive therapeutic targets, which are already partially exploited by novel promising immunotherapies.

## 1. Introduction

Development of incurable metastasis is the cause of the majority of cancer deaths [1]. The bone is an organ frequently colonized by solid tumor metastasis [2]. Statistically, 350,000 people in the United States die each year with bone metastasis. Due to its abundant vascular supply and chemoattractiveness provided by stromal cells, osteoblasts, osteoclasts and osteocytes through production of numerous growth factors and prostaglandins, the bone microenvironment is a breeding ground for the attachment and multiplication of tumor cells [3]. Skeletal metastases are most frequently observed in multiple myeloma (up to 95%), breast and prostate cancer patients (65–80%), while lower rates are observed in patients with lung, kidney, thyroid or other cancers [4,5,6]. In pediatric cancers, bone metastasis is frequent in children suffering from neuroblastoma [7], osteosarcoma [8] and Ewing sarcoma [9].

The evolution of solid tumor metastasis is a complex process. Metastatic dissemination of tumor cells involves pre-metastatic niche formation, tumor cell dissemination through the circulation and chemotactic attraction and homing of tumor cells to the metastatic site of a target organ, as well as reciprocal interactions with local stromal cells and immune cells within the new microenvironment [10,11]. In cases involving prostate cancer metastasis to bone, as an example, it includes at least four steps: 1. colonization (circulating cancer cells enter the bone marrow niche), 2. dormancy (cancer cells adapt to the bone microenvironment and remain dormant), 3. reactivation (cancer cells switch from the dormant state to an actively proliferating state) and 4. reconstruction (cancer cells disrupt the original bone structure and function) [12,13]. These steps have been previously extensively discussed in multiple reviews [10,14,15]. Here, we aim to amalgamate the fragmented knowledge on these aspects that is available for individual cancer entities in order to highlight mechanistic communalities of the bone metastatic process for adult and pediatric solid tumors and consequences for the clinical management of this otherwise fatal condition.

The time at which potentially metastatic cells are released from the primary tumor and move to the secondary site may depend on the tumor type [16]. Several genetic studies and mutational profiling of primary and metastatic tumors suggest that additional genetic events have to occur to enable metastases formation [17]. However, dissemination of certain tumor cells, including Her2-dependent breast cancer, may occur very early, potentially already in the pre-malignant phase of the disease [18], and disseminated tumor cells can remain in a state of metastatic dormancy in colonized tissue for a long time [19]. Apparently, this process is highly ineffective as only a few tumor cells survive in the circulation and home to the metastatic site [20]. Experimental studies show that while up to 80% of tumor cells released from the primary tumor successfully pass the first steps into extravasation, only about 2–4% initiate the growth of micrometastases, and less than 0.01% survive in the new metastatic niche environment and give rise to macrometastases [21,22]. Here, tumor cells continue to evolve but are no longer dependent on the primary tumor. Early divergence from the primary tumor and acquisition of new genetic changes often result in significantly different mutation patterns compared to the predominant clones constituting the primary tumor. This last phase leading to the development of metastasis is very critical and even though there may be a large number of tumor cells in the patient’s blood and tissues, only very few of them manage to establish macrometastasis consistent with the results of experimental studies [23].

For patients, the occurrence of bone metastases is an unfavorable prognostic factor. It worsens their quality of life and increases morbidity and mortality [24,25,26]. In the most common bone metastatic tumors—multiple myeloma, breast and prostate cancer—the expected survival of patients after diagnosis of bone metastases is no longer than two to three years [24]. In part, this is due to bone metastases leading to skeletal morbidity, also referred to as skeletal-related events (SRE), which typically reduce overall survival. Skeletal morbidity is most common in the context of osteolytic bone metastases, which cause pathological fractures, compression of the spinal cord and spinal nerves, pain or neurological deficits, as well as hypercalcemia. In addition, a recent study demonstrated that the bone microenvironment invigorates metastatic seeds for further dissemination by enhancer of zeste homolog 2 (EZH2)-mediated epigenetic reprogramming that confers stem cell-like properties on cancer cells disseminated from bone lesions, thus further negatively impacting patient survival [27].

The main compartment affected by bone metastasis is the red bone marrow, found mainly in the central skeleton, such as the pelvis, sternum, cranium, ribs, vertebrae and scapulae, and to a variable extent in the proximal ends of long bones such as the femur and humerus. Its rich vascularization and unique cellular makeup support the homing of circulating tumor cells and the development of secondary deposits in the bone. Metastases therefore occur predominantly in the axial skeleton (>80% of patients with bone metastases), of which the majority affects the thoracic spine (70%), the lumbosacral region (20%) and cervical vertebrae (10%). Metastases to the pelvic bones, ribs and skull are found in 63, 77 and 35% of cases, respectively. The proximal long bones (humeri and femura) are more frequently affected (53%) than the distal appendicular skeleton (1%) [10].

## 2. Mechanisms of Metastasis

### 2.1. EMT—The First Step towards Formation of Metastasis

Epithelial-to-mesenchymal transition (EMT) is a phenotypic conversion that occurs during embryonic development and tissue remodeling, where epithelial cells obtain mesenchymal-like properties in combination with reduced intercellular adhesion and enhanced motility [28,29]. EMT is a transient and dynamic process that primarily emerges at the onset of tissue invasion. It is tightly controlled by several cellular signaling pathways including epidermal growth factor receptor family tyrosine kinases (ErbB), wingless/integrated (WNT), nuclear factor kappa B (NF-κB) and transforming growth factor (TGF)-β pathways (Figure 1) [30]. The same molecular players are involved in pathological EMT during invasion and metastasis of solid tumors [31]. 

EMT at the basis of metastatic cancer progression is best exemplified by metastatic breast cancer [30]. Breast cancer cell lines with increased in vitro invasiveness and in vivo metastatic potential upregulate the mesenchymal intermediate filament protein vimentin (VIM) [32], reduce cytokeratin levels and components of various cell:cell adhesion complexes such as desmoplakin (DSP), zonula occludens (ZO)-1 and E-cadherin (CDH1), with concomitant reciprocal upregulation of N-cadherin (CDH2) [33,34,35,36]. Thus, CDH2 and VIM are both bona fide markers of breast cancer EMT. Similarly in prostate cancer, markers of a mesenchymal phenotype including CDH2, osteoblast-cadherin (CDH11), and WAP-type four disulfide core/ps20 (WFDC-1) proteins are upregulated correlating with cellular motility, Skp1-cullin 1-F-box (SCF) E3 ligase complex/ubiquitin/proteasome pathway activity and loss of typical prostatic glandular architecture [37,38,39,40,41]. Various factors, such as hepatocyte growth factor (HGF) and epidermal growth factor (EGF) that are altered in the prostate cancer microenvironment through increased production by tumor cells or the cancer-associated stroma are candidates for eliciting EMT [42,43]. The impact of EGF is mediated by caveolae-dependent endocytosis and subsequent transcriptional downregulation of CDH1 by SNAI1. The transcription factor TWIST similarly represses CDH1 expression and upregulates CDH2 in prostate cancer cell lines [44,45]. In contrast, loss of prostate-derived epithelial factor (PDEF), an epithelium-specific ETS transcription factor which is downregulated by TGF-β, induces EMT in PC3 prostate cancer cells [46] as does overexpression of prostate-specific antigen (PSA) and kallikrein-related peptidase 4 (KLK4), both activators of pro-EGF and latent TGF-β2 [47,48]. While PSA and KLK4 are part of normal prostatic secretions, they leak into the tumor microenvironment due to the disruption of the glandular architecture during cancer progression, suggesting a link between tissue architecture and EMT. Further emphasizing the relationship between tumor dedifferentiation and EMT, the hedgehog and bone morphogenic protein (BMP)-7 developmental signaling pathways are reactivated in aggressive prostate cancer and can induce EMT [49]. Of note, BMP-7 is also abundant in the bone microenvironment, potentially contributing to the bone metastatic niche of prostate cancer, as it was shown that prostate cancer cells undergo EMT when experimentally inoculated into mouse bone [50].

### 2.2. Formation of a Pre-Metastatic Niche and Bone Colonization

Current views on tissue tropism of metastasis are based upon specific properties of the metastatic niche. The “seed and soil” hypothesis, articulated by Stephen Paget more than 100 years ago, assumes that a particular tumor cell can settle and begin to proliferate only within a compatible microenvironment [51]. Alternatively, the mechanical hypothesis suggests homing of metastatic tumor cells to be driven by physical capture in tissue capillaries of optimal diameter and blood flow to allow for extravasation at the site of metastasis. With its architecture, bone marrow is an ideal place for such an event. Its sinusoid-shaped capillaries with different throughput, wide gaps between endothelial cells and a thin connective envelope are easily permeable to tumor cells [52,53]. Slow blood flow in the red bone marrow could support the attachment of metastatic tumor cells to the endosteal bone surface. However, the molecular properties of malignant cells (seed) and their reciprocal interactions with the bone microenvironment (soil) are of greater importance in enabling the metastatic spread of the tumor [10,54]. In fact, increasing evidence suggests that primary tumors can prepare the microenvironment of a target organ to create a supportive, pre-metastatic niche for subsequent tumor cell colonization [55]. In this context, it has been shown that cancer-associated fibroblasts (CAFs) in the tumor stroma can prime tumor cells for dissemination to the bone via secretion of C-X-C motif chemokine ligand 12 (CXCL12, also known as SDF-1) [56] (Figure 1). Well in line, overexpression of CXCL12 receptors (CXCR4 and CXCR7) by breast and prostate cancer cells induces chemotaxis along CXCL12 gradients to allow for colonization of the bone [57,58,59,60]. Importantly, high expression of CXCR4 in breast cancer has been associated with a higher incidence of distant metastasis and bone metastasis as compared to low CXCR4 expression [61]. Annexin A2 (ANXA2)/annexin A2 receptor (ANXA2R) interactions between osteoblasts or endothelial cells and circulating prostate cancer cells have also been reported to support homing and adhesion to bone [62,63]. Receptor activator of NF-κB (RANK) expression by prostate cancer cells promotes pre-metastatic niche formation by activating a RANK-L/c-Met-mediated feedforward loop, thus inducing cancer cell colonization to the bone [64]. Tumor-derived proteolytic enzymes such as matrix metalloproteinases (MMPs) can remodel the bone matrix to release and activate growth factors and cytokines, which create a favorable environment for metastatic colonization [65,66,67] (Figure 1). Secretion of lysyl oxidase (LOX) by primary breast cancer cells under hypoxic conditions has been shown to modulate the ECM at bone metastatic sites to create a pre-metastatic niche [68]. Additionally, exosomes and microRNAs produced by the primary tumor cells have been associated with bone remodeling and the development of metastasis in the bone [69,70]. Adhesion molecules such as integrin αvβ3, the receptor for vitronectin, can facilitate the anchorage of disseminating cancer cells to the extracellular matrix (ECM) of the bone niche [71,72]. Furthermore, osteoblast-derived Wnt-induced secreted protein-1 (WISP-1) has been implicated in regulating the adhesion of prostate cancer cells to osteoblasts via VCAM-1/integrin α4β1 (also known as very late antigen-4, VLA-4) [73]. Another study has shown that bone marrow-derived interleukin (IL)-1β promotes metastatic colonization of disseminated breast cancer cells in the bone by stimulating the NF-κB/CREB-Wnt pathway [74]. Finally, binding of metastatic breast cancer cells to E-selectin in the bone vascular niche during early metastatic colonization has been reported to induce mesenchymal-to-epithelial transition (MET), thus facilitating growth at the metastatic site [75,76].

Of note, the precise location and composition of the bone “metastatic niche” of different cancers are poorly defined. It has been proposed to comprise a hematopoietic stem cell (HSC), endosteal (osteoclasts, osteoblasts, osteocytes, fibroblasts) and vascular (endothelial cells, pericytes) niche compartment [13,77,78]. A role in breast cancer colonization of the bone has also been assigned to the adipose tissue compartment in the bone marrow [79]. These niches are considered to be quite stable without undergoing major remodeling over long periods of time, resulting in long latencies in the development of bone metastases [13]. For example, stable microvasculature can retain cancer cells in a dormant state, whereas sprouting vessels activate dormant cells and accelerate micrometastatic outgrowth [80]. Along those lines, long-term quiescent cells such as bone lining cells may thus be more likely to support long-term tumor cell dormancy as compared to bone-synthesizing osteoblasts. Interestingly, human prostate cancer cells have been shown to directly compete with HSCs for occupancy of the endosteal niche during bone marrow transplantation in mice [81]. Most importantly, treatment of mice with HSC-mobilizing agents, such as the CXCR4 antagonist AMD3100 or G-CSF, also resulted in the egression of prostate cancer cells from the bone marrow into the peripheral blood. It therefore seems conceivable that disseminating cancer cells in the bone marrow are subject to similar homing, survival and dormancy mechanisms as HSCs. Breast cancer cells, once nested to bone, undergo partial osteomimicry by acquisition of typical bone cell markers including ICAM-1, CDH11, OPN, osteonectin (SPARC), osteocalcin (BGLAP) and cellular communication network factor 3 (CCN3). This in turn promotes osteoclast-mediated bone resorption through the production of RANK-L, IL-2β, IL-6, IL-11 and tumor necrosis factor (TNF)-α, and supports evasion of an immune response [82,83,84] (Figure 1). Besides breast cancer, osteomimicry has been observed across various bone metastatic cancers including prostate cancer and osteosarcoma [85].

### 2.3. Metastatic Dormancy and Reactivation in the Bone Niche

Solitary disseminated primary tumor cells, once settled in the bone niche, may grow immediately or adopt a nonproliferating dormant state remaining quiescent in the bone marrow for up to decades (“cellular dormancy”) [78,86,87]. A comprehensive mechanistic insight into tumor dormancy, including its implications in tumor invasion and metastasis, has recently been provided [88]. For bone metastases, the fate of colonizing tumor cells most likely depends on their specific location in the bone microenvironment, where around 20% of the endosteal surface undergoes active remodeling, whereas the other 80% remains relatively quiescent at any given time [13]. While actively remodeled surfaces will provide factors that support tumor growth and survival, quiescent compartments of the bone will rather promote dormancy. In the bone microenvironment, both BMP-7 and TGF-β2 have been shown to maintain tumor cell dormancy via reduced focal adhesion kinase (FAK)/EGFR signaling and a high p38 mitogen-activated protein kinase (MAPK) over extracellular signal-regulated kinase (ERK) ratio, favoring cell cycle arrest and inducing a dormant state [89,90].

Specific ligand/receptor interactions with osteoblasts in the bone marrow niche appear to play a major role for the induction of cancer cell dormancy [86,91] (Figure 1). For example, ligand growth arrest-specific 6 protein (GAS6)/AXL interaction with osteoblasts retains bone-metastatic prostate cancer cells in a dormant state [92,93,94]. Interestingly, prostate cancer cells express a repertoire of GAS6 receptors (including AXL, MER and TYRO3), a balanced expression of which has been shown to control dormancy (AXL) or proliferation (TYRO3) [95]. AXL expression is stabilized by hypoxia [96] and hypoxic regions in the bone are less prone to metastasis development [97]. Furthermore, osteoblast-derived growth differentiation factor 10 (GDF10) and TGF-β2 have been reported to induce prostate cancer cell dormancy via activation of TGFBR3–p38 MAPK signaling [98]. Additionally, the WNT5a/receptor tyrosine kinase-like orphan receptor (ROR)-2/Seven in the absentia homolog (SIAH)-2 signaling axis plays a crucial role in inducing and maintaining prostate cancer cell dormancy in the bone [99]. Besides the osteoblast lineage, also other cell populations, such as mesenchymal stem/stromal cells and endothelial cells, have been implicated to play a role in modulating cancer cell dormancy (Figure 1). As such, MSC-derived exosomes have been shown to maintain dormancy of metastatic breast cancer cells and hence to decrease their sensitivity to chemotherapy [100]. Interestingly, breast cancer cells can actually prime MSCs to release microRNA-containing exosomes to promote quiescence [101]. Additionally, secretion of thrombospondin (TSP)-1 from endothelial cells in the perivascular niche can induce dormancy of breast cancer cells [80]. Recently, it has been demonstrated that a subset of periarteriolar MSCs with a neuronal glial antigen 2 (NG2)^+^, Nestin^+^ phenotype produce TGF-β2 and BMP-7 to activate a quiescence pathway in metastatic breast cancer cells by inducing cyclin-dependent kinase inhibitor 1B (CDKN1B) via MAPK activation [102].

Dormant cancer cell reactivation in the bone niche can be influenced by cell-intrinsic changes in gene expression programs in response to various extrinsic cues, including remodeling of the metastatic niche where dormant cancer cells reside, and secretion of tumor-promoting signals [86,87,103,104]. For example, it has been demonstrated that myeloma cells are released from dormancy via RANK-L-driven osteoclastic bone resorption [105]. Accordingly, induction of bone resorption via ovariectomy or castration in mice has been reported to increase the formation of breast and prostate cancer metastases, a process which can be blocked by osteoclast inhibition [20,106,107]. Aberrant VCAM-1 expression by metastatic breast cancer cells has been shown to recruit integrin α4β1^+^ osteoclast precursors, initiating tumor cell reactivation through bone resorption [108]. It seems likely that the frequency of metastases is determined by the number of dormant cells colonizing the endosteal niche as well as the rate of osteoclastic bone resorption, with a higher rate enhancing the chances of dormant tumor cell reactivation. Interestingly, besides their dormancy-promoting function, there is also increasing evidence that osteoblasts contribute to the reactivation of dormant cancer cells in the bone niche (Figure 1). For example, decreased TGF-β and GAS6 expression by osteoblasts can release dormant prostate cancer cells from dormancy [92]. Similarly, reduced secretion of BMP-7 by osteoblasts and MSCs resulted in enhanced prostate cancer cell proliferation in vitro and in vivo [89]. Of note, direct interactions with osteoblasts and cells of the osteogenic niche have been reported to shorten the period of breast cancer cell latency and induce tumor growth in the bone. In particular, heterotypic adherens junctions between breast cancer cell-derived CDH1 and CDH2 in an osteogenic niche can support bone colonization of circulating breast cancer cells and stimulate the mechanistic target of rapamycin (mTOR) pathway-driven micrometastasis formation [109]. Additionally, physical changes in the perivascular niche, such as angiogenic sprouting, can remove suppressive signals (e.g., TSP-1), thus releasing cells from dormancy and triggering cancer cell growth [80]. For example, periostin and TGF-β released from endothelial tip cells of the neovasculature have been shown to promote rapid tumor growth in mice [80].

Upon reawakening, tumor cells undergo MET and start to proliferate. The energy required for growth in the bone niche is at least partly provided by bone marrow-resident adipocytes that release free fatty acids to fuel tumor cell proliferation. They also release leptin (LEP), which stimulates further adipocyte generation from bone marrow MSCs. In addition, LEP has been demonstrated to be involved in tumor cell attraction to the bone marrow niche and to enhance proliferation and migration of tumor cells, while adiponectin from bone marrow adipocytes most likely contributes to tumor dormancy [110] (Figure 1).

### 2.4. Reconstruction: Osteoblastic versus Osteolytic Bone Metastases

Under normal physiological conditions, osteoblast- and osteoclast-mediated bone remodeling is tightly coordinated in time and space, and bone deposition and bone resorption are in balance. However, this balance is disrupted in bone metastatic cancers [104]. Depending on the cancer type, metastatic tumor cell growth in the bone niche either leads to increased production of molecules that disrupt bone homeostasis stimulating osteolysis by osteoclasts, or it causes osteosclerosis through stimulation of osteoblast activity. Although the vast majority of bone metastases of solid tumors present both osteolytic and osteoblastic components, one or the other phenotype prevails in specific bone metastatic cancers.

As an example, the majority of bone metastatic lesions in breast cancer patients is osteolytic [111,112,113]. The process by which breast cancer cells metastasize to the bone is frequently referred to as a “vicious cycle” [114]. It begins when metastatic breast cancer cells produce parathyroid hormone-releasing peptide (PTHrP), which binds to the same receptors as parathyroid hormone and stimulates osteoblasts to secrete increased amounts of RANK-L. The interaction between RANK-L and its receptor RANK on osteoclast precursors stimulates osteoclastogenesis and thus bone resorption. This in turn leads to the release and activation of growth factors such as insulin-like growth factor (IGF)-1, and TGF-β from the bone matrix, which induces cancer cells to proliferate and to produce additional PTHrP (Figure 1). Constitutive osteoclast activation results in sustained osteolysis to provide space for the development of metastatic lesions. Bone degradation also leads to the release of calcium, which further supports the growth of tumor cells expressing extracellular calcium-sensing receptors [115,116]. A variety of tumor-derived factors significantly contribute to the formation of breast cancer metastases in bone, including IL-11, plasminogen activator (PLAU), platelet-derived growth factor (PDGF), fibroblast growth factor (FGF), BMPs and TGF-β [117,118,119,120,121]. Additionally, bone-derived placental growth factor (PGF) is upregulated in the presence of breast cancer cells, resulting in a decreased production of the RANK-L binding glycoprotein osteoprotegerin (OPG) from osteoblasts and stromal cells, thus stimulating the formation of osteolytic lesions due to increased RANK-L levels, which can be prevented by PGF blockade [122] (Figure 1).

Importantly, the vicious cycle model does not take into account the part that dormant cells play in tumor development or the events that initiate the interdependence between tumor cells and osteoclasts. As an alternative to this model, it has been suggested that osteoclasts may not only be bystanders that respond to tumor-derived factors, but rather be responsible for initiating the vicious cycle by first remodeling the endosteal niche, thereby releasing dormant cancer cells from niche-dependent control and reactivating them to form micrometastases [13]. In agreement with this hypothesis, stimulators of osteoclastic bone resorption (e.g., PTHrP overexpression, vitamin D deficiency or calcium restriction) can promote bone metastasis [114,123,124,125], whereas inhibitors of bone resorption such as OPG treatment or bisphosphonates slow down tumor growth in the bone [126,127,128]. Besides their role in the reactivation of dormant tumor cells, osteoclasts may also be involved in fueling the second part of the vicious cycle, when the tumor is already established and starts modifying the microenvironment.

Skeletal metastasis in multiple myeloma predominantly results in osteolytic lesions due to the inhibition of osteoblastic differentiation, reduction in bone deposition and increased osteoclast activity triggered by a variety of osteoclast-activating and osteoblast-inhibitory factors produced by both tumor and stromal cells. These include osteoclast activators macrophage inflammatory protein (MIP)-1α, RANK-L, vascular endothelial growth factor (VEGF), TNF-α, IL-1β, PTHrP, HGF and IL-6. In addition, amphiregulin (AREG) in multiple myeloma-derived exosomes leads to the activation of EGFR in pre-osteoclasts, participating in osteoclastogenesis [129]. Osteoblast inhibition is mediated via direct downregulation of VLA-4/VCAM-1-mediated Runt-related transcription factor 2 (RUNX2) and production of IL-7 as well as the WNT-signaling antagonist dickkopf-1 (DKK-1) [130,131] (Figure 1).

In contrast to bone metastatic breast cancer and multiple myeloma, bone metastatic prostate cancer forms lesions, which are predominantly osteoblastic [132]. This is due to prostate cancer cells preferentially homing to osteoblastic regions of the bone [133]. Unique and direct interactions between bone metastatic prostate cancer cells and osteoblasts provide growth-promoting effects to both cell types (Figure 1). For example, prostate cancer cells secrete BMPs, TGF-β, IGF-1, PDGF, endothelin-1 (ET-1), VEGF and micro-RNA (miR)-940, which promote osteoblast differentiation and activation [134,135,136,137]. This confers a dramatic disruptive effect on the formation of an organized bone structure as well as the collagen matrix, resulting in bone with a disordered spongy structure compared to a compact lamellar architecture. Activated osteoblasts also produce factors that trigger prostate cancer cell proliferation (e.g., VEGF, C-C motif chemokine ligand CCL2, IL-6 and IL-8) [138]. Furthermore, secretion of ET-1 by prostate cancer cells has been shown to suppress DKK-1, activating Wnt signaling and osteoblast-mediated bone deposition [139]. Prostate cancer growth is further enhanced by activated osteoclasts, which release growth factors from the resorbed bone matrix [140]. Prostate cancer-derived exosomes also contribute to extensive osteoblastic lesions delivering ETS1 [141], miR-940 [142] and miR-141-3p [143] to the stroma within the metastatic bone niche. Various survival pathways have been reported to participate in prostate cancer growth in the bone niche, including PTHrP, TGF-β, IGF-1, FGF-2, IL-6 as well as ET-1 signaling [20,144].

Rare osteoblastic and mixed osteoblastic/osteolytic bone lesions can be observed in a fraction of breast cancer patients. These are promoted by communication of tumor cells with osteoblasts and osteoclasts through interaction of variably expressed PDGF and ET-1 with their respective receptors PDGFRα/β and ETAR on osteoblasts and osteoclasts [145,146,147]. Their expression levels determine the development of different types of lesions. In addition, the β-catenin signaling pathway has a significant impact on the bone lesion phenotype and acts as an important determinant in mixed osteoblastic and osteolytic lesions [148]. It was demonstrated that human breast cancer cell lines that preferentially form osteolytic bone metastases exhibited increased levels of Wnt/β-catenin signaling and DKK-1 expression, which blocked Wnt3A-induced osteoblastic differentiation [149].

### 2.5. Bone Microenvironment and Bone Sarcomas

The bone microenvironment is also a fertile soil for the metastasis of the most frequent primary bone sarcomas, osteosarcoma and Ewing sarcoma. Therefore, metastatic bone sarcoma cells can enter, colonize and expand in it, exploiting very similar mechanisms as described above for epithelial cancers and multiple myeloma.

Osteosarcoma is assumed to originate from the bone mesenchymal lineage. Transforming genetic (e.g., *TP53*, *RB* or *CDKN2A* deficiency, aneuploidization), as well as epigenetic aberrations in osteoblast progenitor cells lead to the production of malignant osteoid and immature bone tissue [150,151,152]. In addition, osteosarcoma cells attract normal bone marrow MSCs to the tumor stroma via secretion of CCL2, CXCL1 and TGF-β, supporting their trans-differentiation into cancer-associated fibroblasts (CAFs). As a consequence, CCL2, CXCL1, IL-6 and IL-8 levels are further increased in the tumor microenvironment, eliciting mesenchymal to amoeboid transition of osteosarcoma cells through activation of Ras homology family member A (RHOA) signaling, which promotes motility, invasiveness and trans-endothelial tumor cell migration [153]. Disseminated osteosarcoma cells can resist anoikis (detachment-induced cell death) by expressing fatty acid synthase and potentially related molecules including p-ERK1/2 and BCL-xL [154]. In addition, MSCs support osteosarcoma invasion and metastasis by secreting extracellular vesicles as carriers to transport metastasis-promoting microRNAs (e.g., miRNA-21 and -34a), proteins (e.g., PDGFR-β, tissue inhibitor of metalloproteinases TIMP-1 and TIMP-2), bioactive lipids such as sphingomyelin and metabolites, including glutamic and lactic acids, to the tumor cells [155]. The balance between normal bone formation and bone resorption depends on the acidity of the bone microenvironment. Hypoxia and interstitial acidosis are also important in the promotion of osteosarcoma bone metastasis, in part through activation of IL-8, IL-6, NF-κB1, colony-stimulating factors CSF2 and CSF3, BMP-2, CCL5, CXCL5 and CXCL1 in tumor-associated MSCs [156] (Figure 1). Hypoxia also directly affects migration and invasion of the metastatic tumor cells through induction of hypoxia-inducible factor (HIF)-1α, which is under the control of several factors in osteosarcoma cells including TGF-β1 [157,158], miRNA-20b [159] and miRNA-33b [160]. Besides pH, phosphatidylinositol-3-kinase (PI3K)/Akt is one of the most important signaling nodes that regulates cell motility, adhesion, growth and metastasis of osteosarcoma cells [161,162]. It interacts with MAPK/ERK [163,164], Hedgehog [165,166] and Wnt pathways [167]. Wnt/β-catenin signaling increases the expression of RUNX2 to facilitate metastasis-related gene expression, thus supporting osteosarcoma cell invasion [168,169,170]. Specifically, BMP-2 mediated β-catenin activation and the RhoC/Rho-associated kinase ROCK1/MAPK/Twist1 signaling pathway enhance osteosarcoma growth and promote EMT [171].

In Ewing sarcoma, the pathognomonic driver oncogene EWS-FLI1 modulates several biological pathways including IGF-1, PDGF, VEGF, Wnt and TGF-β signaling, which results in differentiation arrest, proliferation, angiogenesis and immune escape of tumor cells [172]. On the other hand, fluctuations in EWS-FLI1 activity levels promote Ewing sarcoma EMT and increase its metastatic potential [173,174,175]. These may either be due to modulations in the expression of the chimeric protein [175], or the result of perturbations in 3D chromatin organization at EWS-FLI1 binding regions caused by the loss of the cohesion component STAG2 [176,177]. Experimental modulation of EWS-FLI1 upregulates the activity of several key players in EMT, including master regulators SNAI1/SNAI2 and ZEB1, mechanosensitive transcriptional cofactors YAP/TAZ [178] and the NOTCH effector protein and transcriptional repressor Hairy/enhancer-of-split related with YRPW motif protein 1 (HEY1) [179]. Upon entering the circulation, Ewing sarcoma cells overcome anoikis by upregulating IL-1 receptor accessory protein (IL1RAP) enabling metastatic spread [180]. Upon homing to bone, Ewing sarcoma cells produce osteoclast-activating factors, such as IL-6 [181] or TNF-α [182], which induce osteoclast differentiation and activation and thus lead to extensive osteolysis. Additionally, EWS-FLI1 and EZH2, whose mRNAs have been shown to be shed from the tumor cells as an exosome cargo [183,184], may not only bind and block RUNX2 in the tumor cells [185] but also prohibit osteoblast differentiation of mesenchymal cells in the tumor stroma, thus shifting the homeostatic balance in the metastatic bone niche to osteoclast activation. On the other side, when osteoclasts resorb bone, they allow the release of growth factors stored in the bone matrix (IGF-1, TGF-β, PDGF, etc.), which in turn can activate tumor cell proliferation in a vicious cycle [186]. This scenario still needs to be confirmed for metastatic Ewing sarcoma [187].

### 2.6. Bone Metastases in Extraosseous Pediatric Solid Tumors

Among pediatric malignant soft tissue tumors, bone/bone marrow metastasis is rare in rhabdomyosarcoma and retinoblastoma [188,189]. In contrast, the bone marrow is the primary site of metastasis in neuroblastoma, the most common extracranial solid tumor in children. The best-characterized adverse prognostic factor associated with neuroblastoma progression is amplification of the *MYCN* oncogene, which dysregulates protein kinase C (PKC) leading to constitutive phosphorylation of a number of growth factor receptors, and downregulation of CDH2 affecting cell adhesion [190,191]. The combination of MYCN overexpression with caspase-8 depletion, which is frequently observed in human neuroblastoma due to epigenetic silencing, induced the expression of genes involved in EMT and inflammation, and downregulated miR-7a and miR-29b, significantly enhancing bone marrow metastasis in a neuroblastoma mouse model [192]. Overexpression of tropomyosin receptor kinase B (TRKB), the receptor for bone-derived neurotrophic factor, in neuroblastoma cells is associated with increased bone marrow invasion through upregulation of several MMPs (e.g., MMP-1, MMP-2, MMP-3, MMP-9), as well as of serine proteases urokinase and tissue plasminogen activators (PLAU) which degrade the ECM [193] (Figure 1). Co-expression of CXCR4, specifically the 47kD isoform, and CXCR7 significantly and selectively increased neuroblastoma dissemination toward the bone marrow [194]. Bradykinin and ATP leaking from inflammatory processes or damaged cells promoted the attraction of neuroblastoma cells to the bone marrow through stimulating chemokine CXCL12/CXCR4/CXCR7 interactions [195]. Furthermore, CXCR5/CXCL13 and CXCR1/CXCL1 interactions may specifically contribute to neuroblastoma bone marrow metastasis in part by facilitating transmigration of neuroblastoma cells through the bone marrow endothelium [196,197]. Bone marrow infiltration is further supported by IL-6 provided by MSCs upon Galectin-3 stimulation produced from neuroblastoma cells [198]. In turn, the secretome of bone marrow MSCs was shown to promote the expression of the 47 kDa CXCR4 isoform and also increased MMP-9 secretion, expression of integrin α3 and integrin β1, supporting the invasive potential of neuroblastoma cells [199]. A recent study revealed that metastatic neuroblastoma cells alter the frequency and functionality of bone marrow stroma cells, increasing their differentiation capacity towards the osteoblastic lineage, partially mediated by miR-375 shed from the tumor cells in exosomes [200]. In particular, a novel tumor-specific subpopulation of CD146^+^CD271^−^ MSCs characterizes the bone marrow of metastatic neuroblastoma patients [201]. In addition, bone marrow infiltration by neuroblastoma cells increases CD203a and CD73 expression on lymphoid and myeloid cells [202] and derived microvesicles. CD203a—an ectonucleotide-pyrophosphatase-phosphodiesterase—and CD73 catalyze the final steps of immunosuppressive adenosine generation from nicotinamide adenine dinucleotide resulting in inhibition of T-cell proliferation and immune escape [202]. Finally, analysis of neuroblastoma cells isolated from the bone marrow of patients with metastatic disease by immunomagnetic enrichment using anti-GD2 monoclonal antibody identified adaptation to the microenvironment by downmodulation of chemokine (C-X3-C motif) ligand 1 (CX3CL1), angiotensinogen (AGT), Na^+^/K^+^-ATPase alpha 2 (ATP1A2) and upregulation of several genes commonly expressed by various lineages of bone marrow resident cells, such as S100A8 and A9 (calprotectin), CD177, CD3 and CXCL7. Bone marrow-infiltrating neuroblastoma cells also expressed CD271 and human leukocyte antigen (HLA)-G [203]. Together, these data exemplify the mutual reprogramming of the host microenvironment and the infiltrating tumor cells in bone/bone marrow metastasis.

## 3. Diagnosis and Therapy of Bone Metastases

The most common initial clinical symptom of bone metastasis is pain. This is because tumor cells produce substances that irritate unmyelinated C-fiber neurons including bradykinin and substance P. A number of other causes also contribute to the onset of pain, including deformation and consequently, irritation of the periosteum due to small microfractures of the trabecular bone. Advanced stages of bone-metastatic growth are frequently associated with pathological fractures, mainly of the long bones (femur, humerus), which occur predominantly in the proximal parts thereof. In 5–10% of patients with metastatic spinal cord injury, metastases cause vertebral body fractures [204]. The standard-of-care imaging methods to detect bone metastases include X-ray, bone scintigraphy and computed tomography (CT), all of which assess the stromal reaction to cancer cells within the bone marrow, rather than depicting cancer foci themselves [10]. This not only limits early metastasis detection but also the assessment of the treatment response. Therefore, high sensitivity methods including whole body magnetic resonance imaging (MRI) and positron emission tomography-computed tomography (PET-CT) are increasingly used to improve the detection of metastatic bone disease.

At the beginning of bone metastatic growth, the process manifests by osteopenia, detectable by X-ray examination. Pathological changes in bone metabolism and bone structure arising from increased osteoblast activity in the vicinity of metastases is monitored by bone scintigraphy, a radiotracer-based imaging method [205,206,207]. Due to minimal tracer uptake in purely osteolytic lesions, bone scintigraphy is not applicable for the initial diagnosis of patients with multiple myeloma [10]. CT imaging enables the assessment of the condition of cortical and trabecular bone, and a better spatial representation of the skull, pelvis and spine. It also helps to distinguish bone metastasis from hemangioma. However, significant bone destruction or new bone formation must occur before a lesion is detectable in a CT scan [10]. The most sensitive method for assessing bone metastases in the bone marrow is MRI capturing details of 80–90% of bone marrow areas in a single scan [208]. In addition, the recently developed synthetic MRI technique allows the evaluation and quantification of bone lesions, and to discriminate viable progressive osteoblastic from nonviable bone metastases during treatment, as demonstrated for prostate cancer patients [209]. PET-CT uses various isotopes, including the bone turnover-specific tracer fluorine-18 sodium fluoride (^18^F-NaF), as indicators of skeletal metastases and is able to distinguish them from benign lesions [210]. Compared to scintigraphy, PET-CT has a higher sensitivity and specificity, thus enabling the detection of skeletal metastases in early stages [211].

In addition to these imaging technologies, biochemical examinations play an important role in the diagnosis of patients with suspicious bone metastases as has been previously reviewed in detail [212,213]. These include bone isoenzyme of alkaline phosphatase (BALP), BGLAP, carboxy-terminal propeptide of type I procollagen (PICP), N-terminal propeptide of type I collagen (PINP), carboxy-terminal telopeptide of type I collagen (ICTP), fasting urinary pyridinoline (Pyr) and deoxypyridinoline (D-Pyr) collagen degradation products. Serum calcium is also a standard test, and higher calcium levels are an indicator of osteolysis. Bone resorption can be the result of two distinct enzymatic activities. Cathepsin K mediates physiological bone resorption, while degradation by MMP-9 is activated in a pathological situation. ICTP specifically reflects the pathological degradation of bone tissue by MMP-9 and is therefore a highly specific marker of bone resorption caused by a metastatic process.

The therapy of patients with bone metastases is primarily a palliative treatment with the goal of alleviating pain and improving quality of life. Treatment decisions for bone metastases depend mainly on their location, the patient’s general condition and the treatment the patient has received so far. It is usually a combination of local and systemic treatments.

Surgical procedures depend on the size and number of bone metastases and are not applicable for patients with multiple metastatic lesions. During the growth of bone metastases, gradual reduction in bone mass leads to the risk of pathological fractures, mostly of long bones, or collapse of the vertebral body, and compression of the spinal cord or spinal nerves can occur. Such conditions usually require acute orthopedic or neurosurgical treatment. In the majority of cases, there are stabilization procedures for impending pathological fractures, and osteosynthesis or excochleation of a metastatic deposit with filling of the cavity with bone cement. When this is not possible, the limb or spine can be stabilized with an extension, orthosis or corset [214]. In patients with insufficient pain treatment, contra-indications of surgery or risk of progression after surgical stabilization, injection of bone cement or tissue adhesive based on polymethylmethacrylate into the affected area under sciascopic control can be performed using interventional radiology [215,216]. This technique is known as vertebroplasty in cases involving vertebral involvement, or cementoplasty in cases where long bones are affected. These methods provide pain relief in up to 75% of patients. Their advantage is that they do not place too much strain on the patient and can be combined with other therapeutic procedures.

External beam radiotherapy in combination with analgesics is the basic palliative treatment of advanced bone metastatic disease and is a very effective treatment for pain relief. In addition to killing tumor and inflammatory cells preventing discomfort to adjacent nerves, it promotes ossification by destruction of osteoclasts, and thus may stabilize bone. Its advantages are minor side effects, and acting directly against the cause of bone pain. For palliative radiotherapy in patients with bone metastases, the optimal dose-fractionation schedule remains a matter of debate in terms of efficacy, safety and cost-effectiveness. When classical radiotherapy is not practical, radiopharmaceuticals such as radium-223 may be applied instead [217]. Radiopharmaceuticals incorporate into new bone and emit particles (alpha and beta) to kill nearby cancer cells [218,219]. Therefore, they preferentially accumulate in osteoblastic bone metastases, and have been shown to improve both the survival and quality of life of prostate and breast cancer patients [218]. Unfortunately, their application for treatment of osteolytic bone metastasis is limited because of myelosuppression [9].

After local therapy by surgery and/or radiotherapy, systemic neoadjuvant or adjuvant anticancer chemotherapy is applied to patients with bone metastasis. For prostate cancer patients, taxanes (docetaxel and cabazitaxel), which are mitotic inhibitors, provide the only systemic treatment option [220,221,222]. In cases involving breast cancer patients with bone metastases, the chemotherapeutic armamentarium contains several compounds including anthracyclins (inhibitors of DNA synthesis and RNA synthesis) [223], vinorelbine (microtubule disruptor) [224] and capecitabine (thymidylate synthase inhibitor) [225] combined with surgical resection of the primary and metastatic tumors. In the absence of any proven alternatives, chemotherapeutic treatment of osteosarcoma patients with bone metastases still follows the same approach with three agents (methotrexate, cisplatin and doxorubicin) as the treatment of the primary tumor [226]. Similarly, the standard-of-care chemotherapeutic treatment of bone metastatic Ewing sarcoma uses the same backbone regimen as first-line treatment for localized disease, and the benefit of myeloablative dose intensification followed by HSC transplantation remains controversial [227,228].

Analgesic therapy is an integral part of palliative treatment of bone metastases. It is a supportive treatment which, if properly managed, reduces tumor pain to a tolerable level and thus improves the overall condition and quality of life of the patient [229]. For neuropathic and mixed pain, it is advisable to combine analgesics with antidepressants or anticonvulsants. For severe pain, the administration of strong opioids dominates. In addition to the analgesics, corticosteroids may be used to treat pain in multiple skeletal disorders.

### 3.1. Approaches to Biologically Targeted Therapy

The preclinical development of targeted antimetastatic treatments requires adequate models that recapitulate the specific biology of metastases in their niche. For the study of bone metastases, conventional cell line-based or patient-derived xenografts (PDX), which largely retain gene expression patterns and clinical features of the human tumor in rodent hosts, are still considered the gold standard [230,231,232]. However, their use is limited as species-specific factors may prohibit homing of human tumor cells to rodent bones, and the site of tumor cell inoculation greatly impacts organ tropism. An improvement of such systems is provided by subcutaneous human bone implant models, which upon subsequent inoculation of metastatic tumor cells allow them to home to their natural human bone-niche microenvironment in the mouse [233]. Alternatively, an artificial matrix consisting of mature human osteoblasts trapped on a collagen hydroxyapatite substrate, which is then implanted subcutaneously into SCID mice, can be used to generate an easily accessible human bone microenvironment for tumor cell transplantation and anti-bone-metastatic treatment studies [234]. Despite their limitations, xenograft models have been successfully used to unravel the role of tumor-derived exosomes in the bone-metastatic process [235] of bone-derived leukemia inhibitory factor (LIF) and Wnt5 in promoting dormancy of metastatic breast and prostate cancer cells in the bone, respectively [99,236], and of factors promoting bone-metastatic growth including IL-1 in breast cancer [237], G protein-coupled receptor class C group 5 member A (GPRC5A) and NOTCH3 in prostate cancer [238,239]. Xenograft models have proven valuable in the pre-clinical anti-bone-metastatic development of bisphosphonates, the RANK-L-neutralizing monoclonal antibody denosumab, soluble RANK (RANK-Fc), osteoprotegrin-Fc (alone and in combination with docetaxel), several receptor tyrosine kinase inhibitors (dasatinib, saracatinib, and KX2-391) and the α5β3 integrin-targeting antibody etaracizumab [240].

Bisphosphonates and denosumab mechanisms of action have recently been comprehensively reviewed for the treatment of bone metastases [14]. They either directly or indirectly (via inhibition of osteoclast-activating PDGF and VEGF production) inhibit osteoclast activity (Figure 2). They also improve tumor response to treatment by depletion of tumor-associated macrophages that support tumor growth. According to the latest research, bisphosphonates, especially zoledronate, are not only osteoprotective, but also have antiangiogenic, immunomodulatory and antitumor effects. They improve bone metabolism, suppress the formation, growth and spread of bone metastases and have an analgesic effect [241]. While denosumab is currently being tested in combination with bisphosphonates for the treatment of bone metastases in breast and prostate cancer patients [242,243], recent phase III clinical trials adding zoledronic acid to standard backbone chemotherapy of both osteosarcoma and Ewing sarcoma failed to improve patient survival [244,245]. Although bisphosphonates and denosumab inhibit osteoclast activity and tumor-induced osteolysis, they do not restore bone formation. Given the key role of the ubiquitin–proteasome system in controlling the degradation of various bone-related proteins and the importance of regulatory ubiquitination in cancer metastasis [246,247], focus has been drawn to the potential use of proteasome inhibitors for the improvement of bone anabolism [248]. Among these, bortezomib, carfilzomib and ixazomib have shown promising results in stimulating osteoblast differentiation and bone healing through multiple mechanisms. For example, bortezomib can enhance bone formation by increasing the expression of BMP-2 in osteoblastic cells, which leads to increased RUNX2 activity and the upregulation of BGLAP, ALP and collagen I [249,250]. Bortezomib also promotes endoplasmic reticulum stress through IRE1α/XBP1 signaling, resulting in increased expression of various osteoblast markers [251,252]. In mice inoculated with breast cancer cells, bortezomib treatment was shown to reduce osteolytic lesions and to exhibit bone anabolic effects [253]. Bortezomib was further reported to prevent bone metastasis of prostate cancer cells by inhibition of the WW domain containing E3 ubiquitin protein ligase 1 (WWP1) and Smurf ligases, which are frequently upregulated in patients with bone metastasis [254]. Since proteasome inhibitors act in a nonspecific manner, their use as bone anabolic therapy is rather limited. Therefore, conjugation of bortezomib to bisphosphonates as well as the generation of bone-specific nanoparticles have been proposed to selectively deliver proteasome inhibitors to the bone and to prevent systemic side effects [255]. Furthermore, several attempts have been undertaken to specifically target BMPs, PTH or OPG in order to improve bone integrity [256]. Finally, biocompatible polymers may provide novel strategies to enhance bone healing in patients with metastatic bone cancers [257].

For large-scale antimetastatic drug-screening purposes, the use of xenograft models is limited. More recently, bone explant [258] and scalable 3D in vitro models based on organoid and 3D bioprinting technologies have moved into the focus of pre-clinical research. Although they lack vascularization, they at least partially recapitulate the physical, cellular and spatial complexity of the tissue from which they derive, and retain the genetic and functional heterogeneity of the original tumor. As an example, it was shown that organoids generated from human prostate cancer bone metastases retained the same resistance pattern to antiandrogen therapy as the patients’ metastases [259]. An ex vivo tumor/bone co-culture model was used to demonstrate that the CXCL5/CXCR2 axis is sufficient to promote breast cancer colonization during bone metastasis [258]. In addition, the composition of the bioink used in the printing process allows inclusion of bone-specific matrix components. Using this approach, 3D-bioprinted and bone-on-a-chip models for breast cancer bone metastasis [260,261,262] and metastatic neuroblastoma [263] were successfully established.

Among conventional targeted therapies, hormone therapy is used for the palliative treatment of metastatic skeletal disorders in prostate and breast cancer, which depend on sex hormone signaling for growth (Figure 2). In patients with estrogen receptor-dependent breast cancer, antiestrogens (e.g., tamoxifen, fulvestrant) [264,265] and aromatase inhibitors (anastrozole, letrozole, exemestane) are used [266]. Ovariectomy or chemical castration can be performed using analogues of gonadotropin-releasing hormone (luteinizing hormone releasing hormone, LHRH), such as zoladex. In patients with prostate cancer, orchiectomy or treatment with LHRH analogues are primarily performed. If this treatment fails, antiandrogens such as flutamide, which inhibits androgen uptake, or estrogens are administered. Despite antimetastatic benefits for patients, hormone therapy shows several adverse side effects, including osteoporosis [9].

While the therein-mentioned treatment options aim to target established metastases, one way to block metastatic spread already at an early progression step could be to prevent EMT. Therefore, signaling pathways that trigger the EMT process may serve promising therapeutic targets. In part, EMT-associated processes are regulated by TGF-β, EGF and PDGF-β signaling, and inhibitors blocking TGF-β receptor type 1 (e.g., LY364947) [267,268,269] and EGFR type 1 (erlotinib and gefitinib) [270] are currently tested in clinical trials of bone-metastatic disease (Figure 2). Two studies recently described the mechanisms of early dissemination in breast cancer, demonstrating how switching off MAPK and turning on HER2 signaling pathways can activate the EMT process in this cancer [18,271]. Based on the MMTV-Her2 breast cancer mouse model, it was postulated that only a subpopulation of early breast cancer cells with either a Her2^+^, Skp2^high^, Tpl2^low^, phospho-MAPK^low^, CDH1^low^ phenotype, or a Her2^+^, CK8/18^+^, Wnt^high^, phospho-MAPK^low^, Twist1^high^, CDH1^low^ phenotype can disseminate and metastasize. Studies have also underlined a role for MAPKα/β kinases and activating transcription factor 2 (ATF2) in antagonizing Her2 signaling early in cancer progression, and a role of ATF2-mediated blockade of β-catenin activity. In addition, ZEB1, a key player in EMT and the formation of metastatic precursor lesions, was demonstrated to be controlled by miR-1199-5p and miR-200 family members in a double-negative feedback loop [272,273]. Together, these mechanistic findings may open new avenues to target early tumor cell dissemination and prevent bone metastasis in the future.

The formation of the bone metastatic niche and the homing process, which supports the persistence of cancer cells in the new environment, may serve as an alternative therapeutic target. In the homing of breast cancer cells to the bone, E-selectin plays a critical role. By use of the small molecule E-selectin antagonist GMI-1271 (uproleselan) it was shown that inhibition of this molecule significantly prevented entry of breast cancer cells into the bone marrow in pre-clinical models. Therefore, this compound is currently being considered for clinical testing in solid tumors [274] (Figure 2). Integrins critically assist in the formation of the metastatic niche, and molecules such as α5β3 and α4β1 integrins promote the adhesion to ECM components [275]. In addition, ANXA2 and its receptor are critically involved in the adhesion and communication of cancer cells with osteoblasts [63]. Therefore, these factors may also serve as potential targets of therapeutic intervention in the bone metastatic process. Importantly, it has been suggested that homing of disseminated tumor cells to the bone marrow is reversible, thus offering an attractive therapeutic option to potentially kick out these cancer cells from niches where they have become domesticated. Experimentally, plerixafor (a CXCR4 antagonist) has been shown to mobilize disseminated cancer cells from their niches back to the bloodstream [276]. By preventing CXCL12/CXCR4 crosstalk, exposure to plerixafor resulted in reduced ERK1/2 signaling and consequently, decreased proliferation and invasion of disseminated tumor cells (Figure 2).

To target mechanisms responsible for formation of the metastatic niche and of factors activating the dormant state of cancer cells, inhibitors of AXL and TGF-β2 signaling pathways may offer therapeutic options. In cases involving prostate cancer cells, the ANXA2/ANXA2R axis and the GAS6-AXL interaction, which both induce dormancy of cancer cells in the bone microenvironment [277], may represent rational targets for the therapy of skeletal metastasis [278] (Figure 2).

As mentioned earlier, osteoblasts and osteoclasts play important roles in transitions between dormancy and reactivation of disseminated cancer cells. However, mechanisms regulating the timing of dormant tumor cell reactivation leading to metastasis remain an enigma, as do those keeping tumor cells dormant. It is largely unclear how osteoblasts switch from a “dormancy-promoting” to a “metastasis-promoting” state. Recent evidence suggests that bone metastatic breast cancer cells can reprogram osteoblasts in the bone marrow niche to produce altered amounts of decorin and CCN3, which leads to the inhibition of tumor cell proliferation via upregulation of p21 expression. This “tumor-educated” subpopulation of OPN^high^, smooth muscle actin (SMA)^low^, IL-6^low^ osteoblasts may therefore play a critical role in engaging disseminated breast cancer cells into dormancy in early disease [279]. Osteoblasts certainly represent promising candidates for therapeutic targeting to aid in the restriction of metastatic outgrowth in the bone niche. The question remains how one may practically turn this transient dormancy phenotype into a state of permanent metastatic dormancy in a clinical setting.

### 3.2. Potentials for Immunotherapy

Bone marrow represents a unique immune microenvironment containing a complex composition of immune cells that may actually provide an immune-privileged niche for disseminated tumor cells [280]. The diverse interplay between immune cells and the skeletal system has been extensively reviewed elsewhere [281,282,283]. Importantly, tumor cells can create an immunosuppressive bone metastatic microenvironment, which leads to low response rates to different cancer therapies [84]. Briefly, the repertoire of immune cells in the bone marrow niche includes T cells [284], macrophages [285], dendritic cells (DCs) [286], natural killer (NK) cells and myeloid-derived suppressor cells [287]. For example, cytotoxic CD8^+^ T cells can release TNF-α and interferon (IFN)-γ to eliminate tumor cells. The presence of CD4^+^ CD25^high^ regulatory T cells (Tregs) in the tumor and blood of cancer patients predicts poor prognosis and Tregs are significantly increased in the bone marrow of patients with prostate cancer that has metastasized to the bone [288]. Therefore, depleting Tregs in the bone microenvironment may provide a means to prevent bone metastasis.

NK cells represent another important cell type in immune-mediated tumor killing through granzyme B- and perforin-mediated apoptosis or Fas–Fas ligand interactions. Depletion of NK cells causes uncontrolled tumor growth and metastasis. Thus, therapy with NK cells modified to recognize antigens specifically expressed on the surface of cancer cells and to produce cytokines such as IL-2 and IL-15, which increase their survival capacity and proliferation and promote antitumor activity in vivo, provides another treatment option for bone metastasis [289]. The combination of therapy with NK cell-stimulating cytokines (IL-2, IL-12, IL-15 and IL-21), reagents against activities limiting NK cell functionality (anti-KIR/anti-PD1 monoclonal antibodies, Treg depletion) and reagents enhancing tumor cell recognition (monoclonal antibodies, bi/trispecific targeting reagents, chimeric antigen receptors) may hold promise in future NK cell-mediated antimetastatic therapy [290] (Figure 2).

Tumor-associated M2 polarized macrophages (TAMs) promote tumor cell to bone metastasis through CCL2/CCR2 or colony-stimulating factor 1 (CSF1)/CSF1R signaling. Blocking the CCL2/CCLR2 axis can suppress the accumulation of TAMs in tumors as well as reducing metastasis in animal models [291]. Monoclonal antibodies (emactuzumab, cabiralizumab and PD-0360324) and the small molecule pexidartinib (PLX3397) targeting CSF1/CSF1R signaling were demonstrated to reduce the number of TAMs and prevent metastasis in several solid tumors [292,293] (Figure 2). In addition, several compounds including trabectedin, clodronate and zoledronic acid were reported to deplete macrophages by inducing apoptosis [294]. Macrophage polarization may be reprogrammed to tumor cell killing by treatment with tyrosine kinase inhibitors sunitinib and sorafenib, or fenretinide [4-hydroxy (phenyl) retinamide], which inhibit STAT3 or STAT6 in macrophages, thereby blocking IL-10 secretion [295,296].

Dendritic cells suppress the cytotoxic capacity of CD8^+^ T cells via production of various molecules, including arginase I, nitric oxide, TGF-β or IL-10. Dysfunctional DCs can be converted into functional DCs by microtubule destabilizing agents (dolastatin 10 and ansamitocin P3) switching DCs from immunosuppressive to immune-activating by provoking phenotypic and functional DC maturation [297] (Figure 2). Vaccination with tumor antigen-loaded DCs may also provide a means of activating an immune response against bone-metastatic disease [298].

Myeloid-derived suppressor cells (MDSCs) release chemokines including IL-6, VEGF, FGF-2 and MMP-9 to promote cancer progression and bone metastasis. MDSC-targeted therapeutic approaches comprise several modalities: anti-GR-1 antibodies; chemotherapeutic agents (5FU, paclitaxel, gemcitabine, cisplatin, docetaxel and lurbinectedin); phosphodiesterase 5 (PDE5) inhibitors (sildenafil, tadalafil and vardenafil); vemurafenib as well as zoledronic acid, which cause MDSC apoptosis; the mTOR inhibitor rapamycin; STAT3 inhibitors (AG490, CPA7, S3I-201 and stattic) deactivating MDSCs; all-trans-retinoic acid (ATRA) or vitamin D promoting MDSC differentiation into nonsuppressive macrophages and DCs; the COX2 inhibitor celecoxib and nonsteroidal anti-inflammatory drugs (e.g., nitroaspirin); TKIs (sunitinib and sorafenib); antagonists for chemokine receptors (CCR2, CXCR2 and CXCR4) or chemokines (CCL2, CXCL5 and CXCL12) preventing the recruitment of MDSCs from the bone marrow into the tumor microenvironment [280] (Figure 2).

Tumor-associated neutrophils are able to release CXCR4, VEGF and MMP-9 to promote tumor bone metastasis. Targeting any of these immune cells represents a promising avenue for cell-based cancer therapies. This may be achieved by inhibition of CXCR2 or IL-17 to reduce neutrophil migration into the tumor, or by anti-TGF-β that leads to a shift from N2 to the N1 phenotype of neutrophils with subsequent acquisition of antitumor activity [299,300,301] (Figure 2).

Recently, it was shown in mouse tumor models and patients with breast cancer that inhibition of RANK signaling induces an antitumor immune response orchestrated by CD8^+^ T cells. In addition, tumors appear to be more sensitive to anti-programmed death ligand 1 (PD-L1) and/or anti-cytotoxic T lymphocyte-associated protein 4 (CTLA-4) after inhibiting the RANK signaling pathway in tumor cells. This study suggests the use of RANK pathway inhibitors to prime luminal breast cancer for immunotherapy [302]. The TGF-β molecule, which promotes bone metastasis and whose excessive amounts produced from osteoclast-activated tumors completely remodel the bone surface, represents another promising immunotherapy target. Anti-TGF-β treatment was shown to restore functions of Th1 cells, boost immunotherapy and inhibit tumor growth [303].

A number of other immunotherapeutic approaches to fight bone metastasis have been explored. These include chimeric antigen receptor (CAR)-T cell therapy, depletion of Tregs with anti-CD25 antibody (e.g., daclizumab and basiliximab) alone or in combination with cyclophosphamide, fludarabine and paclitaxel-based chemotherapy, or antibodies against immunosuppressive CTLA-4 such as ipilimumab and tremelimumab [280] (Figure 2). In fact, clear evidence that patients with bone metastases would benefit from immune cell-targeted therapies is still lacking. Furthermore, it is important to note that immunotherapies are associated with skeletal-related adverse effects such as compression of the spinal cord, or fractures and lesions caused by increased bone resorption [304,305]. Recently, the concept of osteoimmuno-oncology has been introduced, which takes into account the interactions between tumor, immune and bone cells in the bone microenvironment and may thus provide the basis for the development of more effective immunotherapies against bone metastasis in the future [281].

## 4. Conclusions

Bone metastases are a very common and fatal complication for cancer patients. They worsen the quality of life and are associated with an increase in morbidity and mortality. Established bone metastases are difficult to eradicate with current treatment modalities and increase the risk for further tumor dissemination and progression of treatment-resistant disease. This is due to a complex and reciprocal interplay between tumor cells and the microenvironment in the bone niche. Effective treatments of bone metastatic disease therefore need to block the communication between tumor cells and the various cellular and noncellular components of the host compartment at an early progression stage. To this end, a better understanding of the early steps of metastasis such as pre-metastatic niche formation, escape of tumor cells from the primary site and colonization of the bone is crucial. This may lead to the identification of reliable biomarkers for early diagnosis, which represents a key strategy for the disruption of these early events. In particular, finding ways to prolong or maintain tumor cell dormancy in order to prevent the emergence of overt metastases is an important area for further research. Despite these challenges, advances in our understanding of the bone metastatic process have resulted in the emergence of numerous promising therapies, which target bone cells and/or the bone microenvironment. A recent review provides a comprehensive list of these therapies in current clinical development for the therapy of bone metastases [10]. Furthermore, recent immunotherapeutic developments may hold promise, however, an in-depth knowledge of the immune escape mechanisms utilized by bone metastatic cancers, specifically sarcomas, is urgently warranted. Major attention should also be drawn on the role of the immune microenvironment in controlling disease progression and resistance to therapies. Finally, validation of novel therapeutic approaches requires testing in pre-clinical models of bone metastasis, which are not yet uniformly available for all bone metastatic cancer entities. While genetically engineered animal models for bone metastatic cancer are scarce, novel patient-derived xenograft models, as well as in vitro organoid and scaffold models recapitulating the bone niche, may be increasingly used to accelerate the development of bone metastasis-targeting compounds [259,306,307,308,309,310,311]. This most likely requires a joint and multidisciplinary effort between industry and academia to be eventually successful.

## Figures and Tables

**Figure 1 cells-10-02944-f001:**
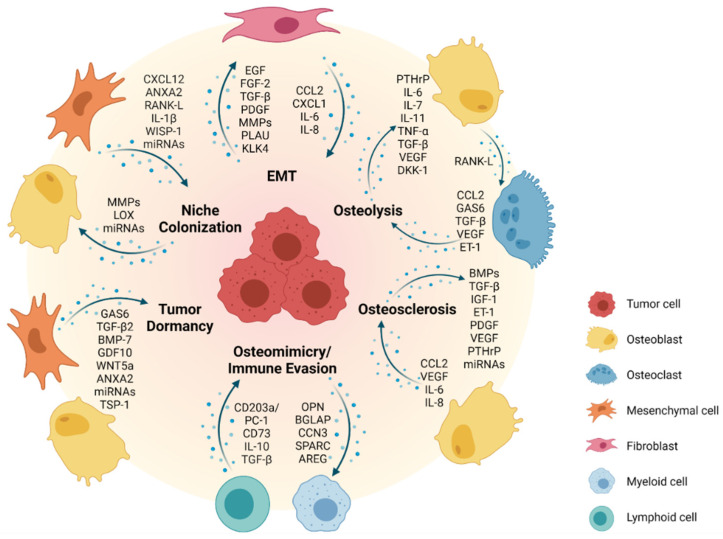
Tumor–microenvironment interactions in bone metastasis. The bone metastatic process involves several steps, including epithelial-to-mesenchymal transition (EMT), colonization of the metastatic niche, tumor dormancy, immune evasion/osteomimicry, bone reconstruction and progression to overt metastases. At later stages, tumor cells secrete factors that stimulate osteoclasts or osteoblasts, leading to excessive bone loss (osteolysis) or bone formation (osteosclerosis), respectively. Most solid tumor metastases to bone exhibit a mixed phenotype of osteolytic and osteoblastic lesions. Highlighted are several key factors that are released by tumor cells and various cell types of the bone metastatic niche mutually controlling their metastasis-driving activities. Created with BioRender.com (accessed on 27 August 2021).

**Figure 2 cells-10-02944-f002:**
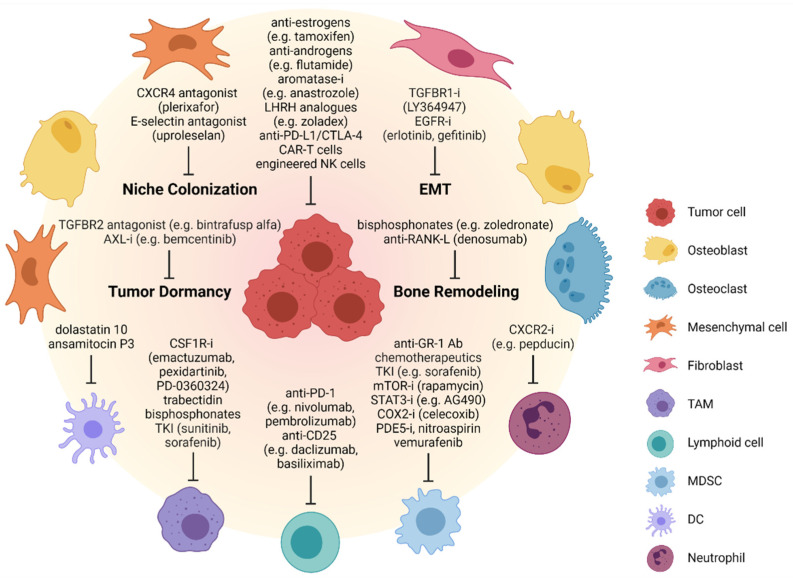
Bone metastasis-targeting drug candidates. Several therapeutic approaches exist to target different steps of the metastatic cascade. For example, the EMT process represents a possible therapeutic intervention point, and inhibitors against TGFBR1 and EGFR1 are currently being tested in clinical trials. Homing and colonization of the metastatic niche by disseminating cancer cells can be targeted using CXCR4 or E-selectin antagonists. Drug candidates targeting dormant tumor cells include AXL and TGFBR2 inhibitors. Bisphosphonates and the anti-RANK-L antibody denosumab can be used to inhibit osteoclast-mediated bone remodeling. Finally, several immunotherapeutic approaches to fight bone metastasis have been explored, including PD-L1 or CTLA-4 checkpoint inhibitors, CAR-T cell therapy and depletion of Tregs with anti-CD25 antibodies. Besides T cells, other immune cell types represent promising candidates for therapeutic targeting, including neutrophils, tumor-associated macrophages (TAMs), lymphoid cells, myeloid-derived suppressor cells (MDSCs) and dendritic cells (DCs). Created with BioRender.com (accessed on 27 August 2021).

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
