# Peer review of "Mechanisms, Diagnosis and Treatment of Bone Metastases"

_cells, 2021, doi:10.3390/cells10112944_

Round 1

Reviewer 1 Report

The manuscript by Ban and colleagues provides an extensive overview on cancer metastases to the bone. It is a state-of-the-art review that covers all relevant aspects of molecular mechanisms and clinical consequences of this type of tumor dissemination. However, the paper somehow lacks stringency by overemphasizing the general aspects of metastases. The authors are therefore encouraged to refer to existing reviews regarding phenomena such as EMT (2.1) and metastatic dormancy (2.3). Additionally, chapter 3 is very descriptive and could be merged with chapter 4 (“Diagnosis and Therapy…”). Here again, the targeted approaches to tackle bone metastases are in part presented in a textbook manner. Especially when it comes to defining targets for immunotherapy, describing the different ways to inhibit specific subtypes of immune cells basically apply to many tumors and are not restricted to treatment of bone metastases, so that references to other reviews will suffice. In turn, the knowledge derived from treating bone metastases in model systems should be focused on. Although these might be only a minority of papers in the field of metastases, this would be more up to the point of this specific topic.

Other minor comments:

Line 55: regarding early dissemination of metastasis: that might be a specific feature of certain tumors including Her2-dependent breast cancer. However, several genetic studies and mutational profiling of primary and metastatic tumors suggest that additional genetic events have to occur to enable metastases formation from primary tumors.

Line 60: here, another review is cited, while the original findings date back to the 1970s e.g. Fidler, JNCI, 1970.

Line 123-124: “correlating with […] pathway”, please clarify what is meant here: pathway activity, enhanced protein turnover?

Line 577 ff The authors discuss and compare the value of ctDNA vs DNA derived from CTCs, however, a link to bone metastases is not presented, further corroborating the idea to omit this part and merge chapters 3 and 4.

Author Response

The paper somehow lacks stringency by overemphasizing the general aspects of metastases.

Response: In the revised manuscript we have reduced the general aspects of metastasis unless their mentioning is immediately relevant to bone metastasis.

The authors are therefore encouraged to refer to existing reviews regarding phenomena such as EMT (2.1) and metastatic dormancy (2.3).

Response: We have shortened the respective paragraphs and refer to relevant reviews.

Additionally, chapter 3 is very descriptive and could be merged with chapter 4 (“Diagnosis and Therapy…”).

Response: Chapters 3 and 4 are now merged in one chapter “Diagnosis and treatment of bone metastases”

Here again, the targeted approaches to tackle bone metastases are in part presented in a textbook manner. Especially when it comes to defining targets for immunotherapy, describing the different ways to inhibit specific subtypes of immune cells basically apply to many tumors and are not restricted to treatment of bone metastases, so that references to other reviews will suffice.

Response: We included references to other recent reviews for the general aspects of immunotherapy. While we agree that the different ways to inhibit subsets of immune cells apply to many tumors, we consider them an important and promising modality in the future treatment of patients with bone metastases. Therefore, we have only slightly shortened this section and modified it to focus on the bone metastatic aspects of immunotherapy.

In turn, the knowledge derived from treating bone metastases in model systems should be focused on. Although these might be only a minority of papers in the field of metastases, this would be more up to the point of this specific topic.

Response: We followed the reviewer´s suggestion and included a paragraph on in vivo and in vitro model systems of bone metastases, and results obtained from their use.

Other minor comments:

Line 55: regarding early dissemination of metastasis: that might be a specific feature of certain tumors including Her2-dependent breast cancer. However, several genetic studies and mutational profiling of primary and metastatic tumors suggest that additional genetic events have to occur to enable metastases formation from primary tumors.

Response: We have changed the text accordingly.

Line 60: here, another review is cited, while the original findings date back to the 1970s e.g. Fidler, JNCI, 1970.

Response: We included the original reference by Fidler et al.

Line 123-124: “correlating with […] pathway”, please clarify what is meant here: pathway activity, enhanced protein turnover?

Response: Done

Line 577 ff The authors discuss and compare the value of ctDNA vs DNA derived from CTCs, however, a link to bone metastases is not presented, further corroborating the idea to omit this part and merge chapters 3 and 4.

Response: In the revised version, we deleted the paragraph on liquid biopsies.

Reviewer 2 Report

General comments:

  1. Suggest could provide a content plan before the introduction section for easy reference.
  2. In the introduction section, need to (i) state earlier reviews close to same topic (if any) and highlight the novelty of current review; (ii) give a brief outline/picture of current review which helps readers to catch up.
  3. 3. Conclusion sections need to further improved, e.g. would be advisable to present some recent studies /attempts with such in vivo human patient studies; and also provide your own suggestions on what can be done as future perspectives.

Specific comments:

  1. In p473-477, authors briefly described the axis of CXCL12/CXCR4/CXCR7 in neuroblastoma invasion. However, in chemokines and corresponding receptors, CXCR4 is known to be important in hematopoietic stem cell homing to the bone marrow and in hematopoietic stem cell quiescence; e.g. studies showed that CXCR4+ breast cancer patients will have their first metastasis near 10 months earlier than CXCR4- patients (Hung et al., Tumour Biol. 2014 Feb;35(2):1581-8). Also, CXCR4+ patients have more bone metastasis than CXCR4-…. Authors should further highlight the physiological roles of CXCR4 in bone metastasis, and discuss the mechanisms underlying.

  1. Intensive studies/recent reviews have showed that posttranslational modifications, e.g. ubiquitination is involved in key mechanisms in tumorigenic microenvironment, carcinogenesis and bone metastasis (CELL CYCLE, 2017, VOL. 16, NO. 7, 634–648; Xuefeng Wu et al., PNAS, 2014, 111 (38), 13870-13875; Molecular Cancer volume 19, 146 (2020); Denise Toscani et a., Appl. Sci. 2021, 11, 4642; BBA-Reviews on Cancer, 2018; 1870(2):165-175….). In this review, authors should consolidate (but not limited to) such information, and give a comprehensive overview by which posttranslational modifications (e.g. UPS) may contribute or perturb bone metastasis via modulating pro-tumorigenic microenvironment. Potential targets as early diagnostic or therapeutic markers should also be proposed.

Author Response

  1. Suggest could provide a content plan before the introduction section for easy reference.

Response: Content plan is included before the Introduction following the advice of this reviewer

  1. In the introduction section, need to (i) state earlier reviews close to same topic (if any) and highlight the novelty of current review; (ii) give a brief outline/picture of current review which helps readers to catch up.

Response: Done

  1. Conclusion sections need to further improved, e.g. would be advisable to present some recent studies /attempts with such in vivo human patient studies; and also provide your own suggestions on what can be done as future perspectives.

Response: Done

Specific comments:

  1. In p473-477, authors briefly described the axis of CXCL12/CXCR4/CXCR7 in neuroblastoma invasion. However, in chemokines and corresponding receptors, CXCR4 is known to be important in hematopoietic stem cell homing to the bone marrow and in hematopoietic stem cell quiescence; e.g. studies showed that CXCR4+ breast cancer patients will have their first metastasis near 10 months earlier than CXCR4- patients (Hung et al., Tumour Biol. 2014 Feb;35(2):1581-8). Also, CXCR4+ patients have more bone metastasis than CXCR4-…. Authors should further highlight the physiological roles of CXCR4 in bone metastasis, and discuss the mechanisms underlying.

Response: We now indicate that high expression of CXCR4 in breast cancer has been associated with early occurrence and a higher incidence of distant metastasis and bone metastasis as compared to low CXCR4 expression. The functional role of the CXCL12/CXCR4 axis as a chemoattractant in metastasis has already been addressed at various points of the review.

  1. Intensive studies/recent reviews have showed that posttranslational modifications, e.g. ubiquitination is involved in key mechanisms in tumorigenic microenvironment, carcinogenesis and bone metastasis (CELL CYCLE, 2017, VOL. 16, NO. 7, 634–648; Xuefeng Wu et al., PNAS, 2014, 111 (38), 13870-13875; Molecular Cancer volume 19, 146 (2020); Denise Toscani et a., Appl. Sci. 2021, 11, 4642; BBA-Reviews on Cancer, 2018; 1870(2):165-175….). In this review, authors should consolidate (but not limited to) such information, and give a comprehensive overview by which posttranslational modifications (e.g. UPS) may contribute or perturb bone metastasis via modulating pro-tumorigenic microenvironment. Potential targets as early diagnostic or therapeutic markers should also be proposed.

Response: We have included a paragraph on ubiquitination and the therapeutic consequences for the use of proteasome inhibitors in the treatment of bone metastases

Round 2

Reviewer 2 Report

This revised version looks good to me.